



# Mixing states of Amazon-basin aerosol particles transported over long distances using transmission electron microscopy

Kouji Adachi[1], Naga Oshima[1], Zhaoheng Gong[2,a], Suzane de Sá[2], Adam P. Bateman[2], Scot T. Martin[2], Joel F. de Brito[3,b], Paulo Artaxo[3], Glauber G. Cirino[4,c], Arthur J. Sedlacek III[5], Peter R. Buseck[6]

[1] Department of Atmosphere, Ocean and Earth System Modelling Research, Meteorological Research Institute, Tsukuba, Japan
[2] School of Engineering and Applied Sciences & Department of Earth and Planetary Sciences, Harvard University, Cambridge, Massachusetts, USA
[3] Institute of Physics, University of São Paulo, São Paulo, Brazil
[4] National Institute for Amazonian Research, Manaus, Amazonas, Brazil
[5] Brookhaven National Laboratory, Upton, New York, USA
[6] School of Earth and Space Exploration & School of Molecular Sciences, Arizona State University, Tempe, Arizona, USA
a: Now at: Harvard Business School, Boston, Massachusetts, USA, b: now at: IMT Lille Douai,
University of Lille, SAGE, Lille 59000, France, c: now at: Geosciences Institute, Federal University of Para, Belem, Brazil.

*Correspondence to*: Kouji Adachi (adachik@mri-jma.go.jp)

**Abstract.** The Amazon basin is important for understanding the global climate both because of its carbon cycle and as a laboratory for obtaining basic knowledge of the continental background atmosphere.
Aerosol particles play an important role in the climate and weather, and knowledge of their compositions and mixing states is necessary to understand their influence on the climate. For this study, we collected aerosol particles from the Amazon basin during the Green Ocean Amazon (GoAmazon2014/5) campaign (February to March 2014) at the T3 site, which locates about 70 km from Manaus, and analyzed using transmission electron microscopy (TEM). TEM has better spatial resolution than other instruments, which
enables us to analyse the occurrences of components that attach to or are embedded within other particles. Based on the TEM results of more than 10,000 particles from several transport events, this study shows the occurrences of individual particles including compositions, size distributions, number fractions, and possible sources of materials that mix with other particles. Aerosol particles during the wet season were from both natural sources such as the Amazon forest, Saharan desert, Atlantic Ocean, and African biomass
burning and anthropogenic sources such as Manaus and local emissions. These particles mix together at an individual particle scale. The number fractions of mineral dust and sea-salt particles increased almost three-fold when long-range transport (LRT) from the African continent occurred. Nearly 20% of mineral dust and primary biological aerosol particles attached sea salts on their surfaces. Sulfates were also internally mixed with sea-salt and mineral dust particles. The TEM element mapping images showed that
several components with sizes of hundreds of nanometres from different sources commonly occur within individual LRT aerosol particles. We conclude that many aerosol particles from natural sources change their compositions by mixing during transport. The compositions and mixing states of these particles after



emission result in changes in their hygroscopic and optical properties and should be considered when assessing their effects on climate.

## 1 Introduction


The Amazon basin can exhibit clean atmospheric conditions at times during the wet season and thus is uniquely useful for understanding aerosol particles from natural background sources (Martin et al., 2010a, 2016). Natural aerosol particles are important for meteorological and climatological influences on this region. Knowledge of background atmospheric conditions is critical for understanding pre-industrial

conditions and, in turn, the climate changes caused by human activity (Andreae 2007; Carslaw et al., 2013; Wang et al., 2016a). To understand the sources, abundances, sizes, and compositions of Amazon aerosol particles, various atmospheric measurements have been conducted such as the Amazonian Aerosol Characterization Experiment 2008 (AMAZE-2008), South American Biomass Burning Analysis (SAMBBA), and the Green Ocean Amazon (GoAmazon2014/5) campaigns (Andreae et al., 2015;

Bateman et al., 2015; Brito et al., 2014; Cirino et al., 2018; Martin et al., 2010a, 2016, 2017). As a result, long-range transport (LRT) from Africa and the Atlantic Ocean was established as an important pathway for mineral dust (hereafter mineral) and sea salt to the Amazon during the wet season (Artaxo et al., 1988, 1990, 1995; Formenti et al., 2001; Krejci et al., 2005; Moran-Zuloaga et al., 2018; Wang et al., 2016b; Worobiec et al., 2007). Biological emissions are also being recognized as possible sources of inorganic-

salt particles containing elements such as K and Na (Pöhlker et al., 2012; China et al., 2018). These particles are mixed within individual particles, and thus, observations of their occurrence at sizes of hundreds of nanometres or smaller are needed to understand their mixing processes in the atmosphere.

Aerosol particles coagulate and can become coated by gas condensation. Such changes affect particle mixing states and compositions (Adachi et al., 2008). The mixing states can significantly affect particle

hygroscopicity, optical properties, and compositions (Adachi et al., 2010, 2011; Adachi and Buseck, 2011; Cappa, et al., 2012; Fraund et al., 2017). It is important to understand and evaluate particle mixing states to accurately simulate the influences of aerosol particles on the climate (Oshima et al., 2009).

Transmission electron microscopy (TEM) is well suited for determining the shapes, mixing states, and compositions of individual particles smaller than 2 μm (Li et al., 2016). TEM measurements have both

advantages and limitations relative to other analytical methods. The advantages of TEM analyses are that they can measure particles smaller than several hundred nanometres, which is the dominant size in terms of particle number concentration. To date, studies have analysed individual particle compositions from samples collected from the Amazon basin using scanning electron microscopy (SEM), scanning transmission X-ray microscopy (STXM), and electron probe X-ray microanalysis (EPMA) (e.g.,

Worobiec et al., 2007; Fraund et al., 2017; Krejci et al., 2005; China et al., 2018; Huffman et al. 2012; Wu et al., 2019). Although each instrument has its advantages and limitations, TEM has better spatial





resolution (~0.1 nm; Li et al. (2016)) than other instruments, which enables us to analyse the shapes and compositions of components with sizes of tens to hundreds of nanometres that attach to other particles. Furthermore, TEM analyses can image the internal structures of individual particles. TEM can also measure refractory materials that are difficult or impossible to detect using other methods, for example, certain types of on-line aerosol mass spectrometry. Limitations of TEM measurement include that it can only be used for particles on substrates, resulting in a loss of volatile material during sampling and analysis, interference from the substrate, and a lower time resolution than that of on-line instruments.

The GoAmazon2014/5 campaign was conducted in 2014 and 2015 and aimed to address questions related to climate processes, perturbations from human activities, and the terrestrial ecosystem in the Amazon basin (Martin et al., 2016). Our study focused on TEM samples collected during intensive observation period (IOP) 1 from 1 February to 31 March, 2014. The goals of this study are to characterize individual aerosol particles collected from the central Amazon basin during the wet season and to measure these particle occurrences and detailed mixing states at an individual-particle level for implications regarding their regional emissions, regional-to-hemispheric scale transport, particle mixing processes, and climate effects.

## 2 Methods

### 2.1 Campaign and sampling

During the GoAmazon2014/5 campaign, atmospheric measurements were conducted at nine research sites and using two aircraft (Martin et al., 2016). We mainly collected TEM samples at the T3 site (3.2133S, 60.5987W), which is 70 km from Manaus and was the most comprehensively instrumented site including the Atmospheric Radiation Measurement (ARM) Mobile Facility One (AMF-1) and the ARM Mobile Aerosol Observing System (MAOS) (Mather and Voyles, 2013). The details of the campaign are described in Martin et al. (2016).

TEM sampling during IOP1 was conducted at the MAOS container using a shared inlet. We used a TEM aerosol sampler (AS-16W, Arios, Tokyo, Japan) with two impactor stages that collect particles with ~0.1-0.7-μm aerodynamic diameter (50% cutoff diameter) on the fine-mode stage and >0.7 μm on the coarse-mode stage. In this study, we used fine-stage samples to focus on fine-mode particles that are abundant in number but have not been analyzed in detail in this area using TEM. The sampler can mount 16 TEM grids (200 mesh Cu grids with lacey carbon substrates (Ted Pella, CA, USA, Type 01881)) attached to an aluminium plate that rotates with a pre-set timer. We used 30-min collection times at a 1.0 L/min flow rate and 2.5 h interval, i.e., eight samples per day from 1 February to 31 March, 2014 (~500 TEM samples total).



## 2.2 TEM analysis

We used a 120 kV transmission electron microscope (JEM-1400, JEOL, Tokyo, Japan) with a scanning
mode (scanning transmission electron microscopy; STEM) equipped with an energy-dispersive X-ray
spectrometer (EDS; X-max 80, Oxford Instruments, Tokyo, Japan). We obtained representative TEM
images of all grid samples (~30 TEM images per sample). Then, 54 samples were selected for
individual particle analyses based on the TEM images to cover the sampling periods with high mineral
particle fractions and other periods of interest (e.g., pollution period and primary biological aerosol
(PBA)-dominant periods). These periods were compared with the model experiments and the
observations using a high-resolution time-of-flight aerosol mass spectrometer (de Sá et al., 2018) and
were classified into LRT, pollution, and background periods (Fig. 1).

Semi-automated particle analysis with 20 seconds of EDS measurement was used for individual particle
analyses. The same measurements yielded particle sizes (area-equivalent diameters) from segmented
binary images. As some particles (e.g., sulfate) can be damaged by an electron beam exposure during
the EDS analysis, we used the STEM images taken before the EDS measurements for the size
measurements. Here, the geometric diameters (area equivalent diameter) can be larger than the
aerodynamic diameters when particles are flat or have low densities, and bouncing from the coarse
stage to the fine stage is possible (Bateman et al., 2017). As a result, although the sampler collects
particles with an aerodynamic diameter of ~0.1-0.7 μm, we have particles up to 3 μm in area-equivalent
diameter. During the STEM-EDS analyses, we analysed all particles larger than 180 nm in two or more
fields of view with magnifications from 5,000 to 12,000 to measure >200 particles per sample. Detailed
information of the analysis is also described in Adachi et al. (2018; 2019) and Ching et al. (2019).

Measured particles were classified into seven categories based on their compositions, i.e., particles
containing both Al and Fe as a proxy of mineral particles, P-bearing particles as a proxy of PBA
particles, Na-bearing particles as a proxy of sea salt and biological salt, S-bearing particles as a proxy of
sulfate, K-bearing particles as a proxy of potassium salt, C-rich particles as a proxy of carbonaceous
particles, and others (Fig. S1). As Na-bearing particles can include both marine and biological sources,
as proposed by China et al. (2018), we use "sea salt" for those having specific Mg and Na ratios
(0.05<Mg/Na<0.3) (Fig. S1). Particles that fall into more than two types (e.g., mineral + sulfate) were
classified into the upper categories in the flow chart presented in Fig. S1. We checked their shapes
individually and confirmed that most particles were classified appropriately. Exceptions that were
categorized as "other" totalled ~3% of all particles and included metal, fly ash, and mineral particles
without Al and Fe (Fig. 1). Nanosphere soot (ns-soot) particles are defined by their aggregated
nanosphere structure and the graphitic-like microstructures observed using TEM (Buseck et al., 2014).
They are grouped into the carbonaceous category because they do not have characteristic elements other
than C.


### 2.3 Model description

We used the Meteorological Research Institute Earth System Model version 2 (MRI-ESM2; Yukimoto et al., 2019) to evaluate the LRT periods and the sources and transport of LRT mineral particles to the Amazon basin (Fig. 2). For this study, we modified the original model configuration and used only atmospheric (the atmospheric general circulation model (AGCM) with land processes (MRI-AGCM3.5)) and aerosol (the Model of Aerosol Species in the Global Atmosphere mark-2

(MASINGAR mk-2)) component models. The model employs horizontal resolutions with an approximately 120-km grid (TL159) and 80 vertical layers from the surface to a model top of 0.01 hPa in a hybrid sigma-pressure coordinate system. The model simulation was performed from January 2008 to December 2015 after a 1-year spin-up run using the prescribed sea surface temperature data (Ishii et al., 2005). In the model simulation, the horizontal wind fields in the AGCM were nudged towards the 6-

hourly Japanese 55-year Reanalysis data (Kobayashi et al. 2015). We used the monthly anthropogenic emissions dataset by Lamarque et al. (2010) and the daily biomass-burning emissions from the global fire assimilation system dataset by Kaiser et al. (2010). Mineral (dust) and sea-salt emissions were calculated depending on the meteorological conditions in the model (Tanaka and Chiba, 2005; Yumimoto et al., 2017).

## 3 Results

### 3.1 Model simulation

The model simulation characterized six LRT periods that have mineral dust concentration peaks greater than 10 μg/m$^3$ during IOP1 (Fig. 2a). The LRT periods cover the high-mineral particle samples of the TEM analysis, and we identify these TEM samples as LRT samples. These periods also generally

overlap those of Moran-Zuloage et al. (2018), who measured aerosol mass concentrations from 1 to 10 μm (coarse mode) at the Amazon Tall Tower Observatory (ATTO) during the campaign, and are consistent with the dust events from the MODIS satellite observations (https://worldview.earthdata.nasa.gov/). The modelled horizontal distributions of mineral dust concentration at the surface level during the six LRT periods and the vertical distributions on 7 March,

when the largest LRT periods occurred during IOP1, are shown in Fig. 2. The model results indicate that these mineral dust particles originated from the Saharan desert, followed by easterly transport in the lower troposphere over the Atlantic Ocean to the Amazon basin (Fig. 2b, c).

### 3.2 Shapes, compositions, and sizes of aerosol particles

Representative particle images with elemental maps (Figs. 3 and S2-10), size distributions, and size-

dependent number fractions for each particle category (Fig. 4) were analysed with TEM. Here, we show the sizes, shapes, and mixing states of individual particles for each particle type.


### 3.2.1 Mineral particles

Tens of Tg per year of mineral particles from the Saharan desert are transported to the Amazon basin
(Graham et al., 2003; Swap et al., 1992; Wang et al., 2016b; Yu et al., 2015) and provide Fe and P

nutrients for forests (Bristow et al., 2010). Such mineral particles also contribute to ice-nucleating
particles (INPs) to form ice clouds (Prennei et al., 2009).

Mineral particles were defined as those containing both Al and Fe, although Na, Si, Ca, S, or K were
also detected in most mineral particles (Figs. 3 and S2). These particles are typically crystalline, appear
darker than other particles, and have irregular edges in TEM images (Fig. 3a). They have a broad size

distribution with a relatively large modal peak (~1.3-1.6 μm in area-equivalent diameter) (Fig. 4a). The
number fractions sporadically increase (Fig. 1a), coordinating with LRT from Africa in the model (Fig.
2).

Most mineral particles consist of Al- and Si-bearing clay minerals (e.g., illite and kaolinite). These
minerals are aggregated with other minerals such as quartz (Si-O), K-feldspar (K-Al-Si), and small

(~100 nm) grains that contain Fe and O or Ca and S (Figs. 3 and S2). The average weight % of Al and
Si in mineral particles in each sample shows a positive correlation for samples from both LRT and other
periods (Fig. 3d). The values for Si/Al between 1.5 and 3 for LRT period samples are roughly consistent
with bulk measurements of Amazon aerosol in other studies (Artaxo et al., 2002; Caquineau et al.,
1998; Martin et al., 2010b; Moran-Zuloaga et al., 2018; Rizzolo et al., 2017). More than half of the

mineral particles are within this Si/Al range, and their possible source is the Saharan desert judging
from our and other model studies (e.g., Moran-Zuloaga et al., 2018). On the other hand, the average
ratios of Si/Al in several non-LRT samples (e.g., the sample from 26 March, 16:11) are lower than those
in the LRT samples (Fig. 3d). This result suggests that the mineralogy and compositions of mineral
particles are different between LRT and other periods and that the latter can have local sources around

the sampling site (e.g., farm and road).

Over half of the mineral particles have Na-bearing particles such as NaCl, $Na_2SO_4$, or their mixtures on
their surfaces. The weight % values of Na-Cl and Na-Mg in the mineral particles show a positive
correlation (Fig. 3e, f), and ~26% of mineral particles from all samples are internally mixed with sea-
salt components. The values of Cl/Na are lower than those in NaCl (Cl/Na ~1.5) because some Cl was

replaced by sulfate, forming $Na_2SO_4$. LRT samples have higher Na, Cl, and Mg fractions than other
samples (Fig. 3e and 3f), suggesting that the LRT particles were mixed with sea salt during transport,
possibly when travelling near the surface of the Atlantic Ocean (Fig. 2c). Lidar measurements observed
similar LRT of dust and biomass burning from Africa in the lowermost 2 km of the atmosphere during
the AMAZE-2008 campaign (Baars et al., 2012). Such mixing of mineral and sea salt in the Amazon

basin is consistent with studies using particle-induced X-ray emission (PIXE) for filter bulk
measurements (Artaxo et al., 1990) or an electron-probe microanalyser and a scanning electron





microscope for individual-particle measurements (e.g., Worobiec et al., 2007; Wu et al., 2019), and we show here that sea salt occurs on the mineral particles as grains several hundred nanometres in size (Figs. 3 and S6).

### 3.2.2 Primary biological aerosol (PBA) particles

PBA particles are important in the Amazon basin because of their abundance and contributions to the regional climate and weather (Martin et al., 2010b; Pöschl et al., 2010). These particles include fungal spores, pollen, bacteria, algae, protozoa, and fragments of plants and organisms (Despres et al., 2012) and are known sources of K, Na, P, Cl, and Fe in the Amazon basin (Artaxo et al., 1993; Krejci et al., 2005; Wu et al., 2019). Biological sources in the Amazon can also emit K and Na salts as fine aerosol particles and influence new particle formation and cloud formation, respectively (China et al., 2016, 2018; Pöhlker et al., 2012), and act as INPs (Prenni et al., 2009).

PBA particles mainly consist of C and O and contain P, S, N, K, Cl, and Na as minor components (Fig. 5). We used P as a tracer for PBA particles to distinguish them from other primary or secondary organic aerosol particles. Elemental distributions of PBA particles show that P occurs either uniformly within particles (Fig. 5; Fig. S3) or as phosphate (Fig. S4). PBA particle shapes depend on the types of biological sources (Fig. 6), although exact source identification requires measurements such as DNA analysis. The size distribution in the PBA particles has two peaks: ~600 nm and >2 µm (Fig. 4). PBA particles >2 µm show unique shapes (Fig. 6). The size distributions and shapes of large PBA particles are consistent with those described by Huffman et al. (2012), who measured PBA particles during the AMAZE-08 campaign. Relative number fractions of PBA particles are higher during non-LRT periods than LRT periods (Figs. 1b and 1c), given that they are emitted within the Basin. PBA number fractions also show diurnal changes; i.e., they are high during the night (3 am in local time) and low during the day (3 pm) (Fig. 5d). The increases in the number fractions of PBA particles during the night can be caused by high relative humidity (RH), which enhances active PBA particle emissions from plants (Elbert et al., 2007; Graham et al., 2003; Huffman et al., 2012; Whitehead et al., 2016). Various PBA particles are mixed with other materials such as minerals, sulfate, Na-bearing particles, or ns-soot (Fig. 5 and S5). For example, PBA particles in Figs. 5 and S5 attach ns-soot, and that in Fig. S3 attaches a mineral particle. The number fractions of internally mixed PBA particles with sea salt are approximately 20% (Fig. 7e). Na occurs on the PBA particle surface as coatings (Fig. S10) or as particle constituents (Fig. 5).

### 3.2.3 Sea-salt particles

Sea-salt particles are emitted from the ocean and act as efficient cloud-condensation nuclei (CCN). Na, Mg, and Cl are commonly used tracers of sea-salt particles, although Cl can be replaced by sulfate and nitrate in the atmosphere (Adachi and Buseck, 2015; Gard et al., 1998; Yoshizue et al., 2019). In this





study, although the sampling site is far from the ocean (>1,000 km), we commonly observed sea-salt particles and their mixtures with mineral particles (Fig. 3 and S6), sulfate (Fig. 7), and PBA particles (Fig. S10). Na-bearing particles, which include both marine and biological sources, have a broad size distribution, with a mode size of ~600 nm (Fig. 4). Number fractions of Na-bearing particles are more

abundant during LRT periods than in other periods (Figs. 1 and 4). The deformed shapes of Na-bearing particles on the substrate suggest that they were hydrated when collected (Fig. 7).

The weight % ratios among Na and Mg within Na-bearing particles show that more than 70% is sea salt that has a Mg/Na weight % between 0.05 and 0.3 (Fig. 7). Some remaining particles can be sea salt because the Mg weight % is lower than the detection limit (~0.1 weight %) within the Na-bearing

particles having relatively low Na weight %. A stoichiometry calculation using S and Cl weight % indicates that ~74 % in mol fractions of Cl was replaced with sulfate. Such internally mixed sea-salt fractions are higher in samples from LRT periods than those in other periods (Fig. 7 d-f).

### 3.2.4 K-bearing particles

Potassium is one of the key elements in the Amazon basin that acts as a seed to form secondary organic

aerosol particles (Pöhlker et al., 2012). Potassium-bearing particles have a modal peak at 400 nm, and their number fractions are similar between LRT periods and other periods (Fig. 4). Many K-bearing particles consist of inorganic salts and include S as sulfate (Fig. S7), C as ns-soot or organic matter (Fig. S7), Cl as chloride (Fig. S5), or their mixtures. In addition, K occurs in mineral dust particles (Figs. 3 and S2) and PBA (Figs. S3 and S4) as a minor component. Potassium is a common constituent of

mineral dust such as K-feldspar and an essential component of biological species and was detected in ~80% of all our analysed particles, mostly as minor fractions.

Pöhlker et al. (2012) found K in nearly all particles and observed that the K mass fraction increased for smaller particles in samples collected at the ATTO site, which has a negligible influence from Manaus. The authors suggested that the K was from biogenic sources but not biomass burning because of a lack

of ns-soot particles. In contrast, our samples include K-bearing particles with ns-soot particles (Fig. S7), and the mixing state is similar to that in biomass burning (Li et al., 2003; Wu et al., 2019; Yokelson et al., 2007). Thus, although there were few wildfires during the wet season, biomass burning from LRT, local biofuel burning, and other anthropogenic sources can contribute to our K-bearing particles collected at the T3 site.

### 270   3.2.5 Sulfate

Nearly 10% of all particles are classified as sulfate (Fig. 1), although S was detected from ~90% of all analysed particles as a minor component of mineral, PBA, and Na- and K-bearing particles. The size distribution of sulfate has a modal peak at ~400 nm (Fig. 4). Nearly 80% of sulfate particles include N, K (<2 weight %), or both, suggesting that they are mostly ammonium or potassium sulfate (Fig. S8).



Anthropogenic pollutants including sulfate from Manaus were sporadically transported to the T3 site (de Sá et al., 2017). For example, during the period that was classified as polluted conditions from Manaus (de Sá et al., 2018), the TEM samples have approximately three times higher sulfate number fractions than those from background periods (Figs. 1 and S8). Sulfate in the Amazon can be formed by the reactions of $SO_2$, dimethyl sulfide (DMS), $H_2S$, or $CS_2$ emitted from local sources (e.g., plants,

microorganisms, and anthropogenic emissions) and LRT sources (e.g., biomass burning and anthropogenic emissions from Africa and the Atlantic Ocean) (Martin et al., 2010b).

### 3.2.6 Carbonaceous particles

Carbonaceous particles include secondary or primary organic aerosol (SOA and POA, respectively), tarball, and ns-soot particles (Fig. S9). The vapor pressure of volatile and semi-volatile organic

compounds was too high to retain them in the particles after sampling and during the TEM analysis, resulting in the loss of the volatile fraction from the particles collected on the TEM grids. Thus, the carbonaceous particle fractions can be underestimated, and we focus only on the non-volatile fractions.

Carbonaceous particles have higher fractions at smaller sizes for both the LRT period and other samples (Fig. 4). SOA particles are organic particles having round shapes with deformation on the substrate when

collected. These particles form through oxidation of low-volatile organic matter emitted from both natural and anthropogenic sources (de Sá et al., 2018). Tarballs originate from biomass burning and have an amorphous structure and spherical shapes on the substrate (Adachi et al., 2019), and their number fractions are <1% in our samples. Ns-soot particles that are internally mixed with K-bearing particles can originate from LRT (Saturno et al., 2018) or local biofuel burning. Other ns-soot particles can be from

anthropogenic sources such as Manaus and neighbouring cities (Glicker et al., 2019). Ns-soot particles are attached to or embedded within the mineral, sulfate, organic, Na- and K-bearing, or PBA particles (Figs. 3, 5, S2, S5, and S7). As ns-soot particles absorb light, they can change the optical properties of these mixed particles into more light-absorbing particles.

### 4 Discussion

Number fractions of aerosol particle types as a function of sizes largely differ between the LRT period and other samples (Fig. 4). At large particle sizes (>2 μm), mineral and PBA particles are dominant during LRT and other periods, respectively, whereas carbonaceous particles dominate in fine particles for both periods. During the LRT periods, most mineral and biomass burning particles such as K-bearing and ns-soot particles could originate from the African continent, and the sea-salt particles came from the Atlantic

Ocean. We assume that these particles were coagulated or condensed during transport (Fig. 8).

Many Na-bearing particles are internally mixed with mineral and PBA particles (Fig. S6) or on the surfaces of biological particles (Fig. S10). In the LRT samples, more than 20% of the mineral and PBA





particles contain sea salt. Sulfate particles are commonly coagulated or condensed with other particles to form internal mixtures. We found correlations of sample average weight % of S between Na-bearing

and mineral particles (Fig. 9). The results suggest that mixing with sulfate occurred during LRT over the Atlantic Ocean and above the Amazon basin rather than at their source regions (Fig. 8).

In the Amazon forest, the RH commonly exceeds >80% during the night. The mixing states of mineral and PBA particles suggest that hygroscopic particles such as NaCl and $(NH_4)_2SO_4$ deliquesce on the surface of these host particles under high RH conditions and thus they increase their sizes by absorbing

water.

The detailed mixing states of individual particles are important to understand their contributions to the climate. In our samples, most particles consist of several components from sources such as the Amazon forest, Saharan desert, Atlantic Ocean, and African biomass burning, some of which are separated by thousands of kilometres (Fig. 8). Although such complex mixing states are common for anthropogenic

aerosol particles (e.g., Ching et al., 2019), we found that particles from natural sources can also be mixed during LRT and form internally mixed particles.

**5 Conclusions**

The compositions and mixing states of individual particles from natural sources in the Amazon basin are important for understanding their roles in cloud formation and radiative forcing as well as for

transport and removal processes. It is difficult to evaluate the compositions and mixing states of refractory particles using on-line instruments. In contrast, the TEM results show that primary particles, including minerals, sea salt, and PBA, dominate in the fine mode particles during the wet season, and their number fractions vary depending on whether they originate from LRT or local sources. This study shows the mixing states of these particles, focusing on the occurrences of individual components on

particles, with sizes ranging from hundreds of nanometres to several micrometres. Sea-salt and sulfate particles commonly occur on the surfaces of mineral and PBA particles. The element mapping images also showed that K and Na occurred either as attached inorganic salts or as constituents of PBA and mineral particles. Our model simulation suggests that mineral particles during LRT periods originated from the Saharan desert. Although mineral particles are originally hydrophobic, the LRT mineral

particles can become hygroscopic by mixing with sea-salt and sulfate particles.

This study focuses on individual particles from samples collected during the wet season in the Amazon basin. Many particles are internally mixed with other components and can have different hygroscopicity and optical properties from their original sources/states. Although previous studies suggested possible local biological sources for K and Na, our observations show that they could originate from sources of

both LRT and local sources. We conclude that many primary particles from natural sources become internally mixed and change compositions while travelling across long distances and various



environments. When mineral particles are mixed with sea salt or sulfate, they can absorb water efficiently, resulting in increasing CCN activity, changing the optical properties, and accelerating their removal from the atmosphere, all of which should be taken into account when considering their actual

occurrence in the atmosphere.

**Data and code availability**

The data sets used in this publication are available at the ARM Climate Research Facility database for the GoAmazon2014/5 experiment (https://www.arm.gov/research/campaigns/amf2014goamazon). The TEM data and the simulation data used in this publication are available upon request (adachik@mri-

jma.go.jp). Access to the MRI-ESM2 code is available under a collaboration framework with MRI.

**Author contributions**

KA and PB planned the TEM experiments. KA carried out the TEM analyses and data processing. KA, ZG, SSdS, APB, JB, GGC, and AJS carried out the TEM sampling and field observations. STM and PA supervised the GoAmazon2014/5 campaign and TEM sampling. NO performed the simulations. KA

prepared the manuscript with contributions from all co-authors.

**Competing interests**

The authors declare that they have no conflict of interest.

**Acknowledgements**

KA and NO thank the support of the Environment Research and Technology Development Fund (5-1605,

2-1703, 2-2003, and 5-2001) of the Environmental Restoration and Conservation Agency of Japan, the Global Environmental Research Coordination System, from the Ministry of the Environment, Japan, and the JSPS KAKENHI (grant numbers JP25740008, JP26701004, JP16K16188, 16H01772, JP18H04134, JP18H03363, JP19H01972, JP19H04236, JP19K21905, and JP19H04259). Institutional support was provided by the Central Office of the Large Scale Biosphere Atmosphere Experiment in Amazonia (LBA),

the National Institute of Amazonian Research (INPA), and Amazonas State University (UEA). We acknowledge the Atmospheric Radiation Measurement (ARM) Climate Research Facility, a user facility of the United States Department of Energy, Office of Science, sponsored by the Office of Biological and Environmental Research, and support from the Atmospheric System Research (ASR) programme of that office. The research was conducted under scientific licenses 001030/2012-4, 001262/2012-2, and

00254/2013-9 of the Brazilian National Council for Scientific and Technological Development (CNPq). We acknowledge the use of imagery from the NASA Worldview application



(https://worldview.earthdata.nasa.gov), part of the NASA Earth Observing System Data and Information System (EOSDIS).

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





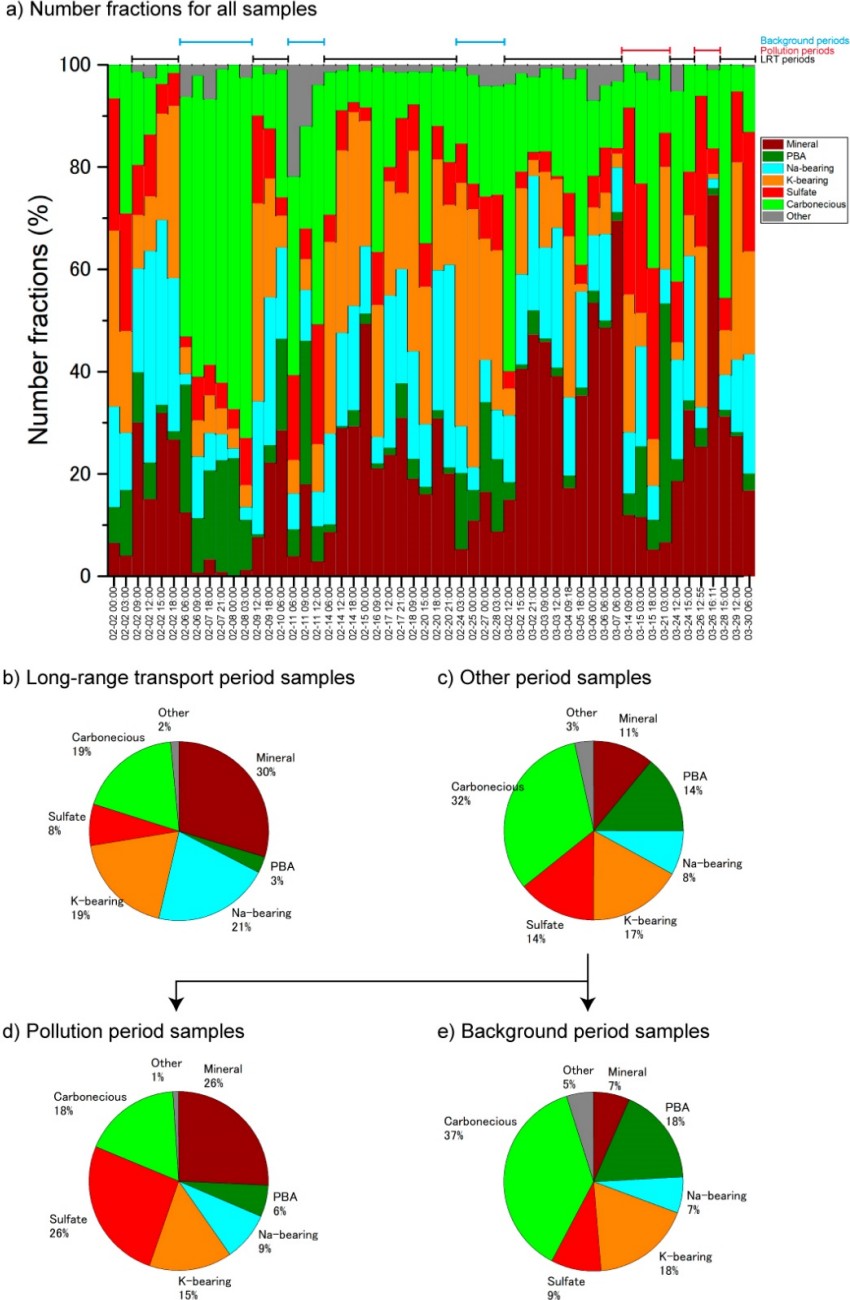

**Figure 1.** Number fractions of each aerosol particle type for (a) all analysed samples and average values during the (b) LRT period and (c) other samples. Other periods were further divided into (d) pollution periods (14, 15, and 26 March, 2014) and (e) background periods (all other periods except the pollution periods and 2 February, 2014) based on the classification by de Sá et al. (2018). Samples collected on 2 February were outside of the measurement period by de Sá et al.

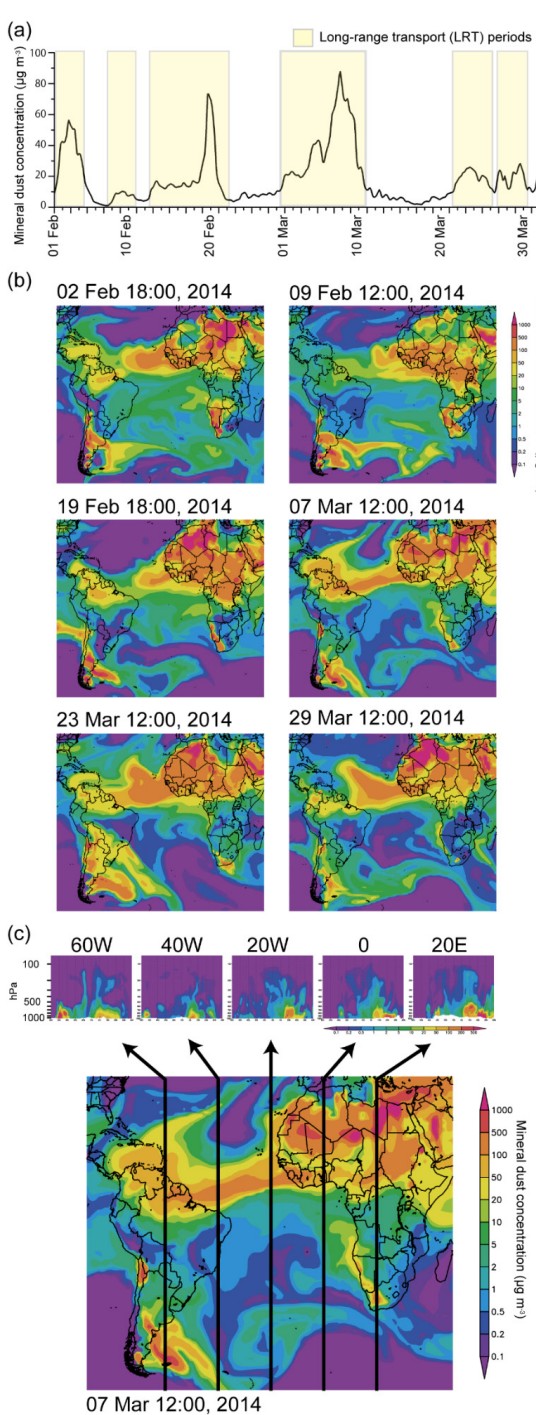


**Figure 2.** Model simulations of (a) temporal variation of mineral dust mass concentrations at the T3 site and snapshots of (b) horizontal distributions of mineral dust concentrations at the surface level for each LRT period and (c) vertical and horizontal distributions in a specific LRT event (7 March, 2014). Yellow screens in (a) show LRT periods.

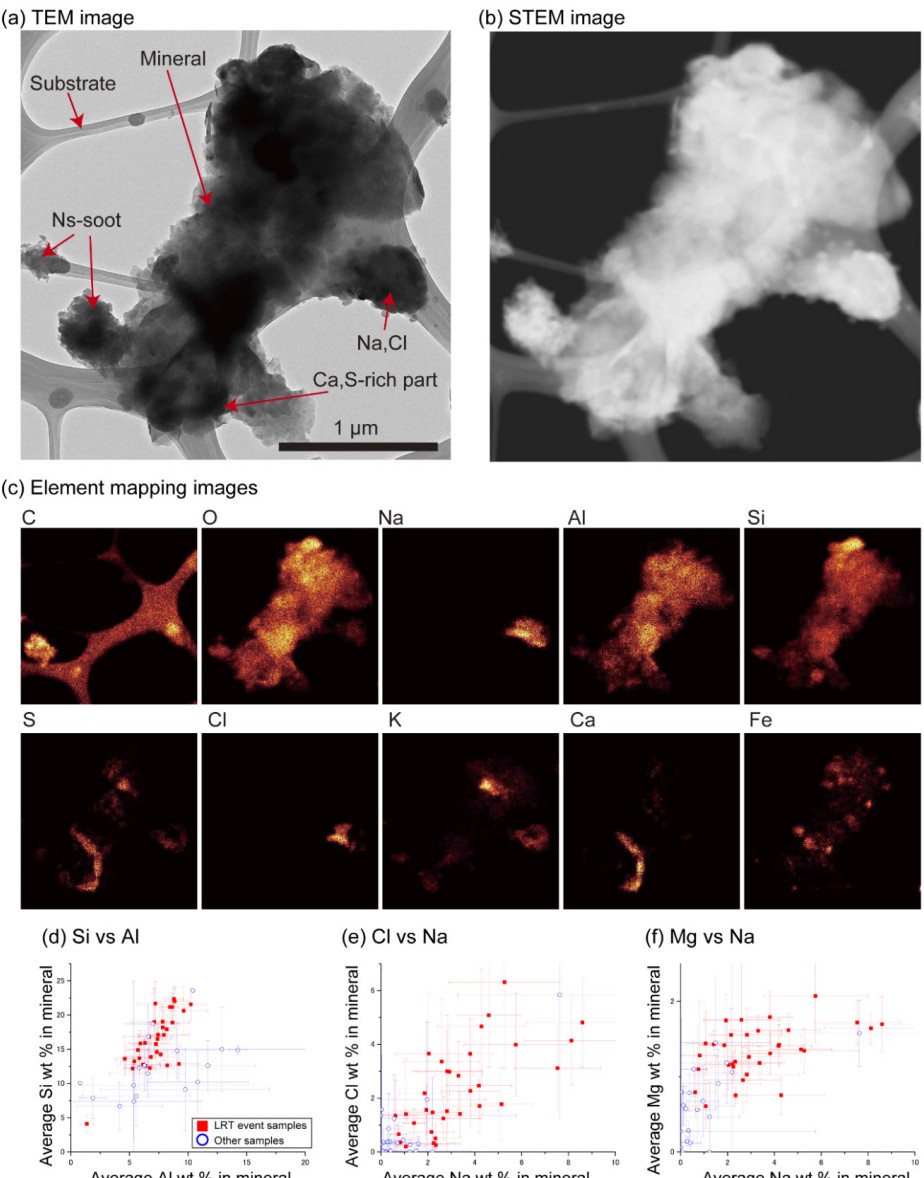


**Figure 3.** Shape and compositions of mineral particles. (a) TEM, (b) STEM, and (c) element mapping images of a mineral particle. This particle was obtained from the sample collected during 15:00-15:30, 2 March, 2014 on a lacey carbon substrate. (d) Relations between average Si and Al weight % in mineral particles, (e) between average Cl and Na weight % in mineral particles, and (f) between average Mg and

Na weight % in mineral particles. Red squares indicate sample averaged values for LRT period samples. Blue open circles indicate sample average values for other (non-LRT period) samples. The sample from 9:00 on 6 February had only one mineral particle and is not shown in the plots because it has a large Al fraction and no Na (0, 30, and 1 weight % for Na, Al, and Si, respectively). Error bars indicate the 95% confidence intervals.



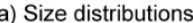

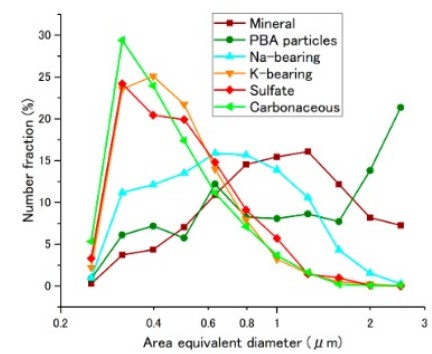

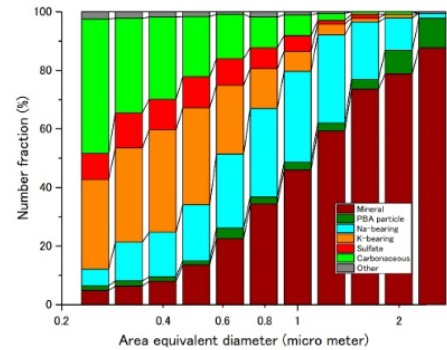

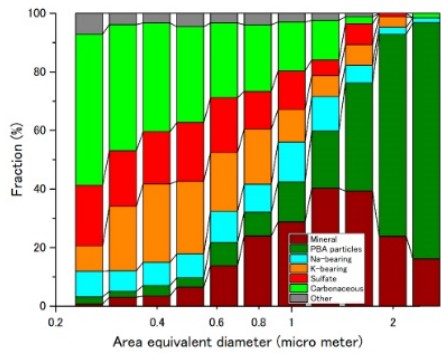


**Figure 4.** Sizes and number fractions of each aerosol type. (a) Size distributions of each aerosol particle type among all samples. The Y-axis shows the number fractions for each bin. N=2519 (mineral), 557 (PBA particle), 1806 (Na-bearing particle), 2223 (K-bearing particle), 1120 (sulfate), 2289 (carbonaceous particle), and 221 (other; not shown). Size-dependent number fractions of each aerosol particle type for

(b) LRT periods and (c) other samples. The size measurements and particle classifications are the same as in Fig. 1. N=7769 (LRT periods) and 2966 (other periods). Size bins are shown on a log scale, and the size ranges are <0.25, 0.25-0.32, 0.32-0.40, 0.40-0.50, 0.50-0.63, 0.63-0.79, 0.79-1.00, 1.00-1.26, 1.26-1.58, 1.58-2.00, and >2.00 μm.





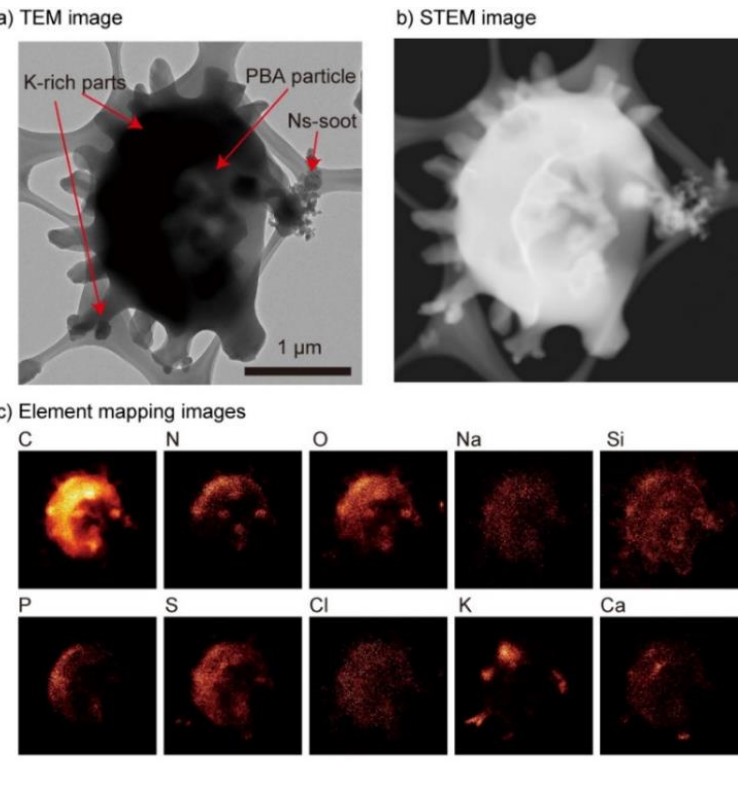

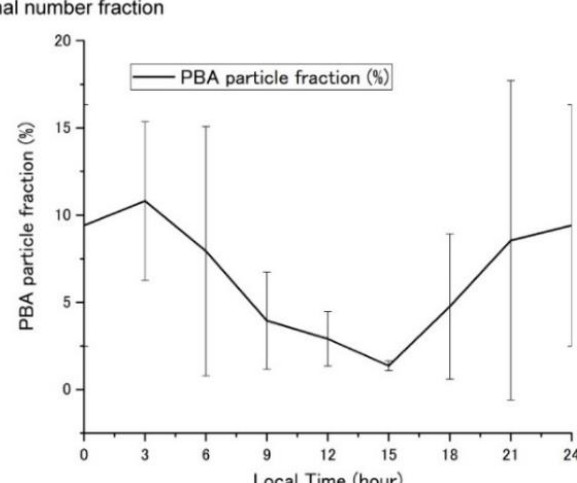

**Figure 5.** Shape, compositions, and diurnal abundance of PBA particles. (a) TEM, (b) STEM, and (c) element mapping images of a PBA particle. This particle contains K-rich and C-N-O-rich grains and attaches an ns-soot particle. The sample was collected from 0:00-0:30, 27 February, 2014. (d) Diurnal number fractions of PBA particles. The fractions were averaged values for all samples. Error bars indicate the 95% confidence intervals.


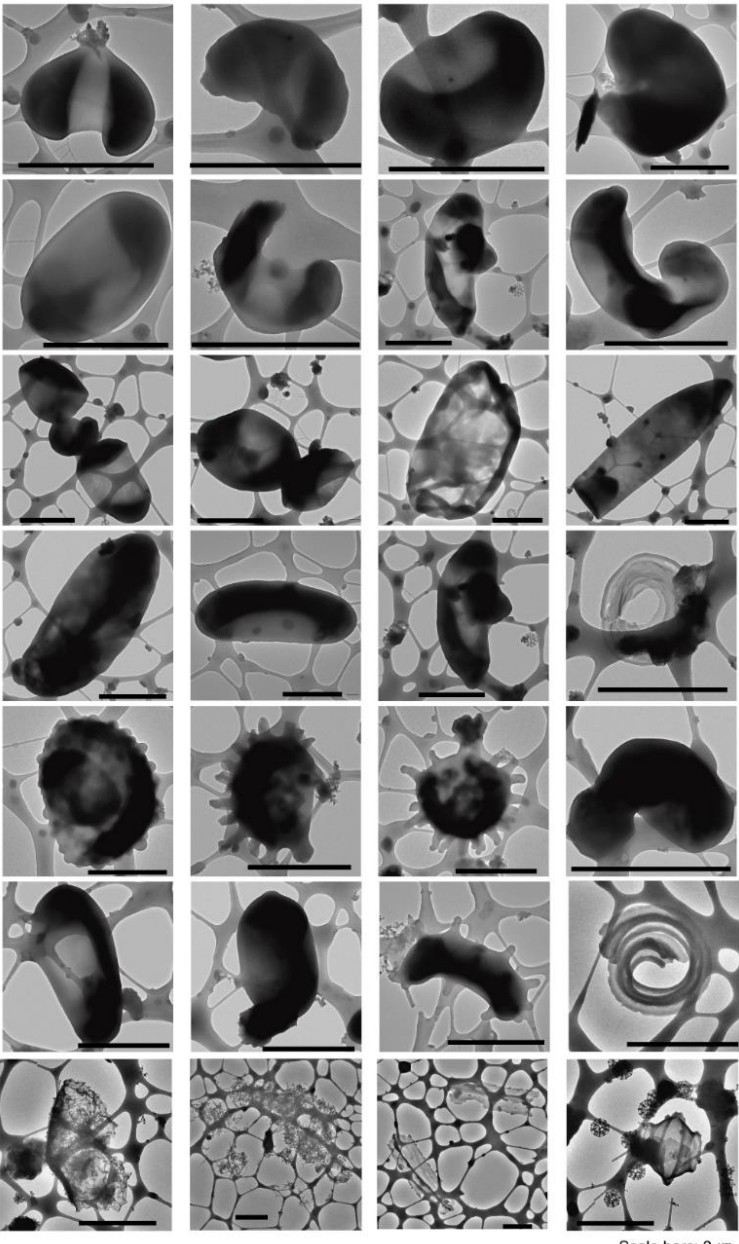

Scale bars: 2 μm

**Figure 6.** Examples of TEM images of PBA and the relevant particles. The TEM images were obtained from samples collected during other (background) periods (09:00, 7 February; 15:00, 10 February; 0:00, 27 February; 3:00, 15 March, 2014; and 09:00, 18 March, 2014). Brochosome particles, which originate from leafhoppers (Wittmaack, 2005), were abundant in some samples (e.g., bottom right image) and were

included in the "other" category because they do not contain P. Some particles were too fragile to analyse the compositions (e.g., three images from the bottom left image), although their shapes imply that they have biological sources.



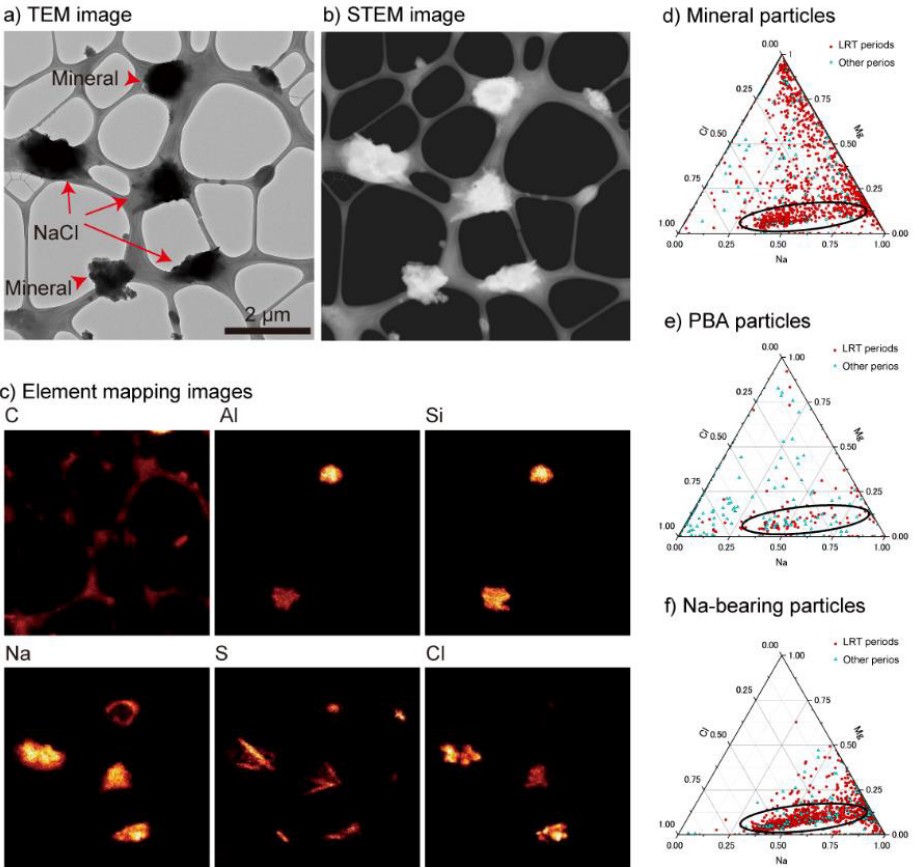

**Figure 7.** Shapes and compositions of sea-salt particles. (a) TEM, (b) STEM, and (c) element mapping images of sea-salt and mineral particles. These sea-salt particles also contain S as sulfate. The sample was collected from 12:00-12:30, 2 February, 2014. Weight % ratios among Na, Mg, and Cl within (d) mineral particles, (e) PBA particles, and (f) sea-salt particles. The black ellipse areas indicate typical sea-salt compositions from a marine source.


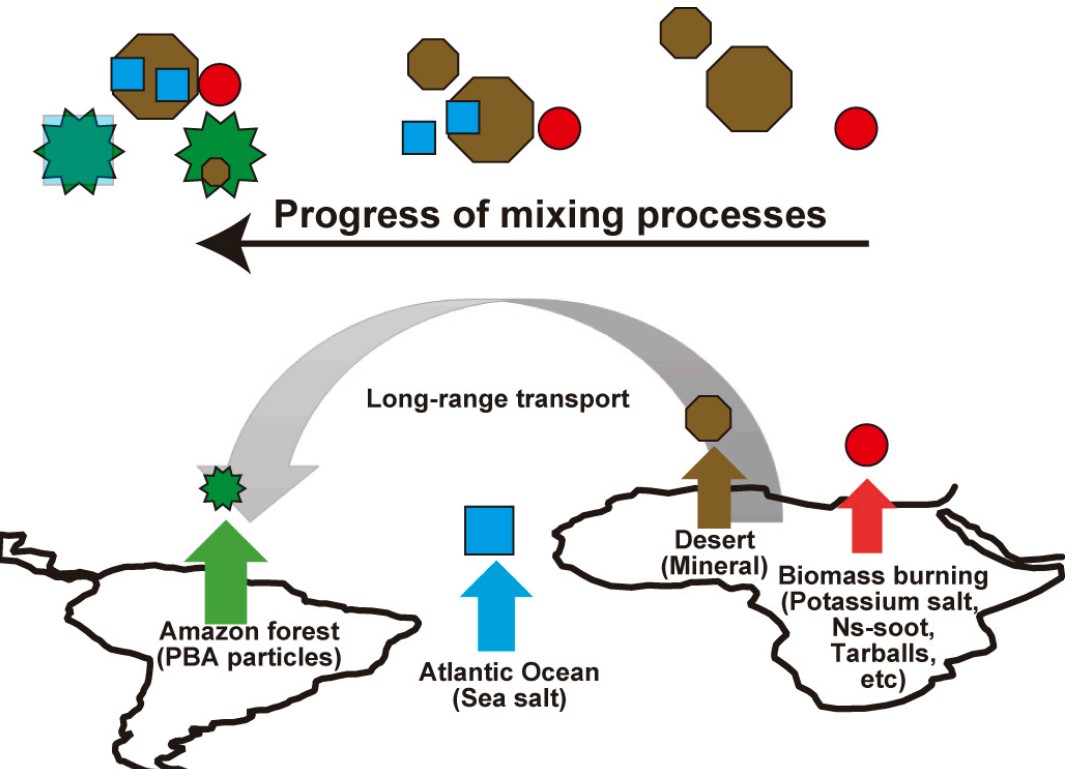

**Figure 8.** Schematic image of possible particle sources and transport pathways. Top images show the progress of individual-particle mixings during LRT. The bottom image shows possible major sources for each particle (Martin et al., 2010b).

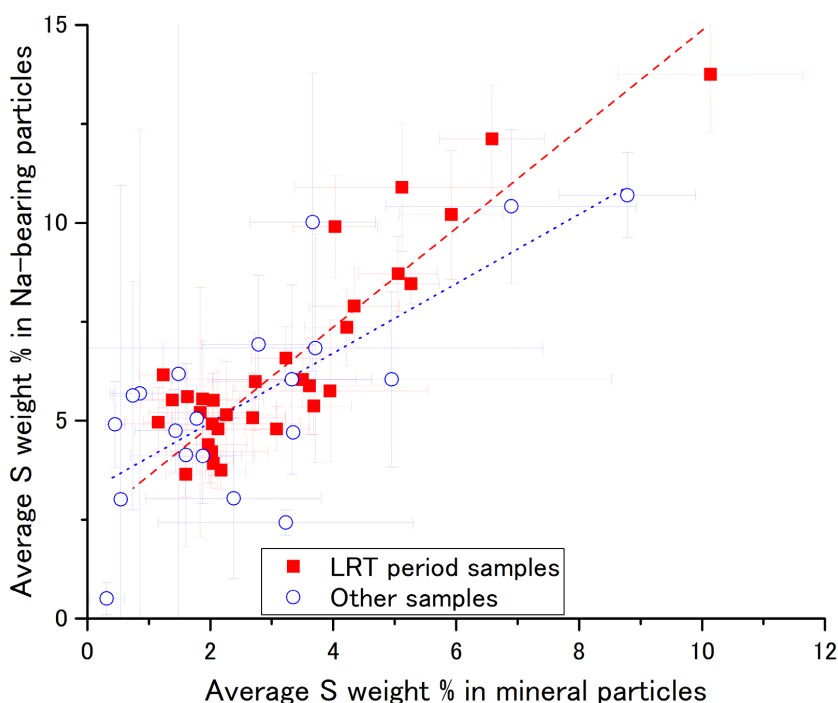

**Figure 9.** The relation between average S weight % in sea-salt and mineral particles within LRT periods
(open red squares) and other (blue circles) samples. Coefficients $R^2$ of determination were 0.80 and 0.51
for the LRT period (red dashed line) and other samples (blue dotted line), respectively.