# Peer review of "Mixing states of Amazon-basin aerosol particles transported over long distances using transmission electron microscopy"

_Atmospheric Chemistry and Physics, 2020_

## Referee Comment (RC1) · Anonymous Referee #1 · 1 Jul 2020

In this study, composition of Amazon-basin aerosol particles has been studied using transmission electron microscopy (TEM). Aerosol particles were collected during the Green Ocean Amazon campaign in 2014 and over 10000 particles were analyzed. TEM has a very high spatial resolution, which allows the analysis of the composition of individual particles in very detail. However, it has some disadvantages such as particle evaporation during analysis. The results showed that particles have different composition, size distribution and number fractions based on they origin and the particles were mixed to more complex structures during their transport in the atmosphere. In general, aerosol particles play an important role in atmospheric processes and the composition of particles is necessary to know when they influence on the climate is studied. Thus,

the study is important and topical.

To my knowledge, this kind of detailed study on individual particles by TEM has not been done before from Amazon-basin particles. Furthermore, it reveals novel information of particle composition, origin and transport. The used methods are very suitable for analysis of composition of individual particles. The manuscript (MS) is quite well organized and written. The content on the MS is in the scope of journal. Thus, the MS is suitable for publication in this journal. However, I have some comments, suggestions and technical corrections that should be considered and discussed before publication.

Main comment:

The main conclusion is that many aerosol particles change their composition by mixing during transport. TEM results showed that analyzed particles are often mixtures of particles from different origins (and composition) indicating that particles have been coagulated in the atmosphere during the transport. How have you ensured that the particles have not just agglomerated on the TEM grid during collection? Have you estimated how many particles per unit area you can collected onto TEM grid in order to avoid particles agglomeration during sampling (i.e. the particles are separately on TEM grid)? Please estimate particle agglomeration on TEM grid and clarify this in the revised version of MS.

Specific comments/technical corrections:

Structure of the MS is now: 1 Introduction, 2 Methods, 3. Results, 4 Discussion, and 5 Conclusions. The structure where Results and Discussion are combined into one chapter could be clearer. Now Results chapter also contains some discussion.

Line 186: . . .shows a positive correlation => Please indicate correlation coefficient/R2.

Line 197-198: . . .show a positive correlation => Please indicate correlation coefficient/R2.

Line 238-239: Na, Mg, and Cl are commonly used tracers of sea-salt particles. . . =>

Why element map for Mg is not shown in the figures?

Line 245-246: The deformed shapes of Na-bearing particles on the substrate suggest that they were hydrated when collected (Fig. 7). => Is it possible that particles are already hydrated before collection in the atmosphere? Please clarify.

Line 322: ... form internally mixed particles => Is the impact of internally mixed particle of climate different than that of externally mixed particle?

Line 330-331: Sea-salt and sulfate particles commonly occur on the surfaces of mineral and PBA particles. => Is the simple reason that they are just wetted in the atmosphere (or during sampling)? Please clarify in the MS.

References:

Authors of the MS have many own references in the reference list (e.g. Adachi 7 the first author references, Artaxo 5). Please check that all references are needed and remove unnecessary references.

Figures:

Figure 1. The smallest fonts are too small, practically texts are not readable (legend texts, x-axis labels, periods text). Please increase font size (double it in the smallest ones).

Figure 2. Too small fonts. Color bars too small (especially (c)). Yellow shaded area is too light to see, especially in print (a).

Figure 3. Data points and error bars not clear in (d)-(f). Fonts of numbers too small. Please plot 1:1, 1:2 or 1:n lines show assumed relations (or trendlines with R2 values).

Figure 4. Too small fonts (especially legend texts) and unclear plots. Please make clearer plots. Sulfates, carbonaceous and K-bearing have the highest values at smaller particles. Are they formed via new particle formation? What is reason?

[Figure]

Figure 5. Plot d) is unclear.

Figure 6. Does water vapor change the shape of PBA particles (see e.g. 2nd image form bottom/left)? Please clarify.

Figure 7. Plots d)-f) are unclear (too small fonts).

Supporting Figure 3a, 4a and 5a. Please indicate acronym PBOA (PBA?).

---

## Referee Comment (RC2) · Anonymous Referee #2 · 4 Jul 2020

The study works on individual aerosol particles collected in Amazon basin where is important for carbon cycle as the green ocean in the world. The authors collected the aerosol particles during the Green Ocean Amazon compaign at T3 site. They used TEM to analyzed different kinds of particles during the LRT particles. They carefully divided particle types based on their compositions and mixing states. Also, they accounted for their number fractions and compared them during LRT and non-LRT periods in the Amazons. Certainly, the study is critical important to understand properties of background aerosols and LRT particles in the Amazons. Finally, they found these particles are internally mixed particles, in particular, sulfate and sea salts as important coatings. Therefore, I think that the knowledge is filled into the gaps in the Amazon.

[Figure]

Before the paper accepted, I would like have some minor comments here.

(1) I might suggest that the authors need to add clear TEM images as the Figure 1 including different types of particles, such as K-bearing, Sulfate, soot, and mineral particles..Then you can give the Figure showing their number fractions. The potential readers might be know better clear the particle types.

(2) It is better to add more discussion. I noticed that they didn't deploy more discussion in discussion part. I might suggest that you need to consider the result from LRT and non-LRT. What are they different? And how could they have large impacts on ecology or climate? Also, the authors can add discussions on mixing structure of different aerosol components. These could be have more literature and scientific discussion should be enhanced.

Some minor comments:

L58: coated by acidic gas condensation

L165 during the IOP1 (Figure 2)

L200, Did you observe the NaNO3? NaCl can easly react with HNO3 and H2SO4 forming NaNO3 and Na2SO4.

L223, you observed the 600 nm and >2 um. Recently, Li et al., (2020) found two modes, suggesting different types of biological particles. I am wondering whether the authors observe observed more types with different morphologies. Here the PBA name can be changed to PBAP. PBAP can be more acceptable in many studies such(Després et al., 2012); (Pöhlker et al., 2012); (Li et al., 2020).

L227-229,The diurnal variations can commonly occur in the forests in the worlds. The emission mechanisms could be very complicated depending on winds, RH, and Temperature on plants and soil (Li et al., 2020)(Elbert et al., 2007).

L251, Again, what about nitrate.

[Figure]

L270, Here the authors found 10% sulfate. Maybe need to add one TEM image to show their morphology in Figure 1?

L289, Maybe add one TEM image showing the SOA and POA? As the authors found many sulfate particles. The SOA is internally mixed with sulfate as its coating? For example, Li et al. (2016) shows most of SOA coat on secondary inorganic particles in the forest and remote air.

L294, why did the authors used biofuel not biomass here?

L308 other aerosol components

L309 form various internal mixtures. As Li et al., raise the concept models of mixing structure. The sulfate can be core with SOA coating. They also can be coating on mineral, POA or others.

References: Després, V., Huffman, J. A., Burrows, S. M., Hoose, C., Safatov, A., Buryak, G., Fröhlich-Nowoisky, J., Elbert, W., Andreae, M., Pöschl, U., and Jaenicke, R.: Primary biological aerosol particles in the atmosphere: a review, Tellus B, 64, 15598, 10.3402/tellusb.v64i0.15598, 2012. Elbert, W., Taylor, P. E., Andreae, M. O., and Pöschl, U.: Contribution of fungi to primary biogenic aerosols in the atmosphere: wet and dry discharged spores, carbohydrates, and inorganic ions, Atmos. Chem. Phys., 7, 4569-4588, 10.5194/acp-7-4569-2007, 2007. Li, W., Sun, J., Xu, L., Shi, Z., Riemer, N., Sun, Y., Fu, P., Zhang, J., Lin, Y., Wang, X., Shao, L., Chen, J., Zhang, X., Wang, Z., and Wang, W.: A conceptual framework for mixing structures in individual aerosol particles, J. Geophys. Res., 121, 13,784-713,798, 2016. Li, W., Liu, L., Xu, L., Zhang, J., Yuan, Q., Ding, X., Hu, W., Fu, P., and Zhang, D.: Overview of primary biological aerosol particles from a Chinese boreal forest: Insight into morphology, size, and mixing state at microscopic scale, Science of The Total Environment, 719, 137520, https://doi.org/10.1016/j.scitotenv.2020.137520, 2020. Pöhlker, C., Wiedemann, K. T., Sinha, B., Shiraiwa, M., Gunthe, S. S., Smith, M., Su, H., Artaxo, P., Chen, Q., Cheng, Y., Elbert, W., Gilles, M. K., Kilcoyne, A. L. D., Moffet, R. C., Weigand, M., Martin,

S. T., Pöschl, U., and Andreae, M. O.: Biogenic potassium salt particles as seeds for secondary organic aerosol in the Amazon, Science, 337, 1075-1078, 2012.

---

## Author Comment (AC1) · 24 Aug 2020

Response to the reviewer 1 comments for "Mixing states of Amazon-basin aerosol particles transported over long distances using transmission electron microscopy" by Adachi et al.

Referee comments are shown in black, normal font.
**Authors replies are shown in red bold font.**
***Revised texts are shown in red, bold and italic font.***

Anonymous Referee #1

In this study, composition of Amazon-basin aerosol particles has been studied using transmission electron microscopy (TEM). Aerosol particles were collected during the Green Ocean Amazon campaign in 2014 and over 10000 particles were analyzed. TEM has a very high spatial resolution, which allows the analysis of the composition of individual particles in very detail. However, it has some disadvantages such as particle evaporation during analysis. The results showed that particles have different composition, size distribution and number fractions based on they origin and the particles were mixed to more complex structures during their transport in the atmosphere. In general, aerosol particles play an important role in atmospheric processes and the composition of particles is necessary to know when they influence on the climate is studied. Thus, the study is important and topical. To my knowledge, this kind of detailed study on individual particles by TEM has not been done before from Amazon-basin particles. Furthermore, it reveals novel information of particle composition, origin and transport. The used methods are very suitable for analysis of composition of individual particles. The manuscript (MS) is quite well organized and written. The content on the MS is in the scope of journal. Thus, the MS is suitable for publication in this journal. However, I have some comments, suggestions and technical corrections that should be considered and discussed before publication.

**Authors reply: We appreciate the reviewer's comments to improve our manuscript. We considered all reviewer's comments and revised our manuscript based on the suggestions.**

1-1. Main comment:
The main conclusion is that many aerosol particles change their composition by mixing during transport. TEM results showed that analyzed particles are often mixtures of particles from different origins (and composition) indicating that particles have been coagulated in the atmosphere during the transport. How have you ensured that the particles have not just agglomerated on the TEM grid during collection? Have you estimated how many particles per unit area you can collected onto TEM grid in order to avoid particles agglomeration during

sampling (i.e. the particles are separately on TEM grid)? Please estimate particle agglomeration on TEM grid and clarify this in the revised version of MS.

The particle concentrations on the substrates in the Amazon samples were generally much lower than those in urban plume samples (e.g., Tokyo, Japan; Adachi et al. 2016) because of the clean environment in the Amazon basin. We chose 30 min sampling time with 1 L/min flow rate, and the sampling condition worked well for collecting particles with not too sparse but not having many agglomerations. The medium number concentration is 0.16 particle per $\mu m^2$, and the total particle area over the analyzed area is 7%. The results indicate that most particles are dispersed on the substrates. We also avoided areas that having overloaded particles, i.e., the area just below the impactor nozzle tends to have too many particles. Comparing to uniform substrates (e.g., formvar substrates), lacey carbon substrates prevent liquid particles from spreading over the substrate and forming mixed particles on the substrate. Nevertheless, although we minimized the possibility to having agglomerated particles on the substrates, it is still possible that some particles were overlapped on the substrate when collected. If such particles occurred, we would overestimate the number fractions of internally mixed particles. Based on the particle area over the analyzed area (7%), we estimated the uncertainty is at most 10% and mentioned the value in the revised text.

*The lacey carbon substrates minimize interference from the substrate during the particle composition measurements and a spread of liquid particles over the substrate. We used 30-min collection times at a 1.0 L/min flow rate to have appropriate particle number concentration on the substrates.*

*We randomly chose two to seven areas that do not have too many particles with a magnification of 6,000 (23×23 $\mu$m) to avoid particles that overlap on the substrate. Each field of view includes ~74 particles on average, resulting in ~7% of particle area among the field of views. As a result, we reduced the chance to analyze particles that agglomerate on the substrate. Nevertheless, it is still possible that some particles overlapped on the substrates. We evaluate that the overestimation for the number fraction measurements of internally mixed particles is less than 10% based on the particle area fractions.*

Specific comments/technical corrections:
1-2. Structure of the MS is now: 1 Introduction, 2 Methods, 3. Results, 4 Discussion, and 5 Conclusions. The structure where Results and Discussion are combined into one chapter could be clearer. Now Results chapter also contains some discussion.

The sections "3 Results" and "4 Discussion" are combined to "3 Results and discussion". The discussion section is revised to "3.3 Possible mixing processes and implications to the climate"

1-3. Line 186: . . .shows a positive correlation => Please indicate correlation coefficient/R2.
1-4. Line 197-198: . . .show a positive correlation => Please indicate correlation coefficient/R2.

$R^2$ values were added to the caption.

*Coefficients $R^2$ of determination for (a), (b), and (c) were 0.60, 0.47, and 0.24 for the LRT period samples and 0.18, 0.66, and 0.40 for other samples, respectively. Trend lines are shown in red and blue broken lines for LRT period and other samples, respectively. The correlation between Si and Al for other samples (a) is relatively low because of possible contributions from two sources (LRT and local dusts).*

1-5. Line 238-239: Na, Mg, and Cl are commonly used tracers of sea-salt particles. . . => Why element map for Mg is not shown in the figures?

Mg-mapping images were added in the revised Figure 11. A lack of figure space was the reason for not having the Mg mapping image in the original figures, and thus we divided the figure into two.

[Figure]

Figure 11. Shapes and compositions of sea-salt particles. (a) TEM, (b) STEM, and (c) element mapping images of sea-salt and mineral particles. These sea-salt particles also contain S as sulfate. The sample was collected from 12:00-12:30, 2 February, 2014.

1-6. Line 245-246: The deformed shapes of Na-bearing particles on the substrate suggest that they were hydrated when collected (Fig. 7). => Is it possible that particles are already hydrated before collection in the atmosphere? Please clarify.

Yes. The text was revised.
*The deformed shapes of Na-bearing particles on the substrate suggest that they were hydrated in the atmosphere and on the substrates when collected (Fig. 11).*

1-7. Line 322: . . . form internally mixed particles => Is the impact of internally mixed particle of climate different than that of externally mixed particle?

Yes. We mentioned the effect in this section and the conclusion.
*Although such complex mixing states are common for anthropogenic aerosol particles (e.g., Ching et al., 2019), we found that particles from natural sources can also be mixed during LRT and form internally mixed particles, resulting in different hygroscopicity and optical properties.*
*Conclusion:*
*When mineral particles or other hydrophilic-primary particles are mixed with sea salt or sulfate, they can absorb water efficiently, resulting in increasing CCN activity, changing the optical properties, and accelerating their removal from the atmosphere, all of which should be taken into account when considering their actual occurrence in the atmosphere to evaluate their climate influences.*

1-8. Line 330-331: Sea-salt and sulfate particles commonly occur on the surfaces of mineral and PBA particles. => Is the simple reason that they are just wetted in the atmosphere (or during sampling)? Please clarify in the MS.

We think that there are two possible processes to form the mixtures. First process is coagulation, which is a process that, for example, sulfate particles collide with mineral dust particles in the atmosphere. Both hydrated and non-hydrated sulfate (and sea salt) can coagulate with mineral-dust particles. Second possible mixing process is condensation. Gases $H_2SO_4$, for example, condenses on the mineral-dust surface. Hydrated particles can spread over a substrate and form an internal mixture when they directly deposit on the post-deposit particles on the substrate. However, the process is not the main process to form the mixed particles, as mentioned in our reply in 1-1. As the LRT particles have more processed mixed particles than other background particles, we interpret that the mixings mainly occurred

during the LRT.

*As the LRT particles have more processed mixed particles than other background particles, we interpret that the mixings mainly occurred during the LRT through coagulation, condensation, or both.*

References:
1-9. Authors of the MS have many own references in the reference list (e.g. Adachi 7 the first author references, Artaxo 5). Please check that all references are needed and remove unnecessary references.

We removed Adachi and Buseck (2011) and Artaxo et al. (1988) from the reference list as the references overlap with others.

Figures:
The authors' general comments for figures: The original figures were partially unclear because of low resolutions in the pdf file (due to the size limit of the pdf file). We hope the problem will be clarified when published on ACP. We also divided some figures into two so that we can have more space for each panel.

1-10. Figure 1. The smallest fonts are too small, practically texts are not readable (legend texts, x-axis labels, periods text). Please increase font size (double it in the smallest ones).

The figure 1 was revised and divided into two figures (Figs. 1 and 2) to have a space for large texts and additional TEM images.

1-11. Figure 2. Too small fonts. Color bars too small (especially (c)). Yellow shaded area is too light to see, especially in print (a).

Revised. The figure was divided into two figures (Figs. 3 and 4) to have a space for large texts and color bars.

1-12. Figure 3. Data points and error bars not clear in (d)-(f). Fonts of numbers too small. Please plot 1:1, 1:2 or 1:n lines show assumed relations (or trendlines with R2 values).

The figure was revised and divided into two figures (Figs. 6 and 7) to have a space for large texts and an additional TEM image (Mg). Trend and 1:n lines are added. $R^2$ values were also

added to the caption.

[Figure]

*Figure 7. Relations (a) between average Si and Al weight % in mineral particles, (b) between average Cl and Na weight % in mineral particles, and (c) between average Mg and Na weight % in mineral particles. Red squares indicate sample averaged values for LRT period samples. Blue open circles indicate sample average values for other (non-LRT period) samples. The sample from 9:00 on 6 February had only one mineral particle and is not shown in the plots because it has a large Al fraction and no Na (0, 30, and 1 weight % for Na, Al, and Si, respectively). Coefficients $R^2$ of determination for (a), (b), and (c) were 0.60, 0.47, and 0.24 for the LRT period samples and 0.18, 0.66, and 0.40 for other samples, respectively. Trend lines are shown in red and blue broken lines for LRT period and other samples, respectively. The correlation between Si and Al for other samples (a) is relatively low because of possible contributions from two sources (LRT and local dusts). Error bars indicate the 95% confidence intervals.*

1-13. Figure 4. Too small fonts (especially legend texts) and unclear plots. Please make clearer plots. Sulfates, carbonaceous and K-bearing have the highest values at smaller particles. Are they formed via new particle formation? What is reason?

The figure was revised. Carbonaceous and sulfate particles have the highest number fraction at the smallest size bin, although K-bearing particles have the highest number fraction at ~400 nm (the third bin). The size distributions in our study generally agree with that in Pöschl et al. (2010). New particle formation hardly occurs near the ground in the Amazon basin (e.g., Pöhlker et al., 2012) but can happen in the lower free troposphere (Wang et al., 2016). Particles from new particle formation are transported to the boundary layer, and condensation on such preexisting particles can be a pass to form these particles.

*Wang et al. (2016a) showed that small particles are transported from the free troposphere into the boundary layer by vertical transports during precipitation events, and such particles can be nuclei of the SOA particles in the background condition.*

*The size-dependent number fractions of samples from other period are roughly consistent with the results during the AMAZE-08 campaign (Pöschl et al., 2010).*

References:

Pöhlker, C. et al.: Biogenic Potassium Salt Particles as Seeds for Secondary Organic Aerosol in the Amazon, Science, 337, 1075-1078, 10.1126/science.1223264, 2012.

Pöschl, U. et al..: Rainforest Aerosols as Biogenic Nuclei of Clouds and Precipitation in the Amazon, Science, 329, 1513-1516, 10.1126/science.1191056, 2010.

Wang, J., et al.: Amazon boundary layer aerosol concentration sustained by vertical transport during rainfall, Nature, 539, 416-419, 10.1038/nature19819, 2016a.

1-14. Figure 5. Plot d) is unclear.

The figure was revised.

1-15. Figure 6. Does water vapor change the shape of PBA particles (see e.g. 2nd image form bottom/left)? Please clarify.

It is possible. PBA particles can absorb water and change their shapes. We added the following sentence in the caption.

*As these PBA particles were observed under vacuum, they can have inflated shapes when observed in an ambient condition.*

1-16. Figure 7. Plots d)-f) are unclear (too small fonts).

The figure was revised and divided into two figures (Figs. 10 and 11) to have a space for large texts

1-17. Supporting Figure 3a, 4a and 5a. Please indicate acronym PBOA (PBA?).

PBOA should be PBA. The figures were corrected.

---

## Author Comment (AC2) · 24 Aug 2020

Response to the reviewer 2 comments for "Mixing states of Amazon-basin aerosol particles transported over long distances using transmission electron microscopy" by Adachi et al.

Referee comments are shown in black, normal font.

**Authors replies are shown in red bold font.**

*Revised texts are shown in red, bold and italic font.*

Anonymous Referee #2

The study works on individual aerosol particles collected in Amazon basin where is important for carbon cycle as the green ocean in the world. The authors collected the aerosol particles during the Green Ocean Amazon compaign at T3 site. They used TEM to analyzed different kinds of particles during the LRT particles. They carefully divided particle types based on their compositions and mixing states. Also, they accounted for their number fractions and compared them during LRT and non-LRT periods in the Amazons. Certainly, the study is critical important to understand properties of background aerosols and LRT particles in the Amazons. Finally, they found these particles are internally mixed particles, in particular, sulfate and sea salts as important coatings. Therefore, I think that the knowledge is filled into the gaps in the Amazon.

Before the paper accepted, I would like have some minor comments here.

**Authors reply: We appreciate the reviewer's comments to improve our manuscript. We considered all reviewer's comments and revised our manuscript based on the suggestions.**

2-1. (1) I might suggest that the authors need to add clear TEM images as the Figure 1 including different types of particles, such as K-bearing, Sulfate, soot, and mineral particles. Then you can give the Figure showing their number fractions. The potential readers might be know better clear the particle types.

**We added representative TEM images of each particle type in the revised Fig. 1.**

[Figure]

Figure 1. Number fractions of each aerosol particle type for all analysed samples (a) and the representative TEM images of each particle type (b). Scale bars for Mineral, PBA, Na-bearing, and sulfate are 1 μm, and those for K-bearing and carbonaceous are 200 nm. Compositions and low-magnification images of these particles are shown in Fig. S2 (mineral), Fig. S5 (PBA), Fig. 11 (Na-bearing), Fig. S7 (K-bearing), Fig S8 (sulfate), and Fig. S7 (carbonaceous). Samples were divided into LRT and other periods (bars in the panel (a)). Other periods were further divided into pollution periods (14, 15, and 26 March, 2014) and background periods

*(all other periods except the pollution periods and 2 February, 2014) based on the classification by de Sá et al. (2018). Samples collected on 2 February were outside of the measurement period by de Sá et al.*

2-2. (2) It is better to add more discussion. I noticed that they didn't deploy more discussion in discussion part. I might suggest that you need to consider the result from LRT and non-LRT. What are they different? And how could they have large impacts on ecology or climate? Also, the authors can add discussions on mixing structure of different aerosol components. These could be have more literature and scientific discussion should be enhanced.

We combined the results and discussion sections in the revised manuscript (please also see our reply to 1-2). Now we have only "results and discussion" section and have more discussion than the original based on the reviewers' suggestions.

Examples of additional discussions.
*The size-dependent number fractions of samples from other period are roughly consistent with the results during the AMAZE-08 campaign (Pöschl et al., 2010).*
*Nitrate can also react with NaCl and form sodium nitrate. However, we did not observe it in this study because of lower concentrations of nitrate than sulfate during the campaign (de Sá et al., 2018) and insensitivity of nitrate detection in the current TEM analysis.*
*Wang et al. (2016a) showed that small particles are transported from the free troposphere into the boundary layer by vertical transports during precipitation events, and such particles can be nuclei of the SOA particles in the background condition.*
*As the LRT particles have more processed mixed particles than other background particles, we interpret that the mixings mainly occurred during the LRT through coagulation, condensation, or both.*

Some minor comments:
2-3. L58: coated by acidic gas condensation

Revised as follows.
*Aerosol particles coagulate and can become coated by volatile organic compounds or acid gas condensation.*

2-4. L165 during the IOP1 (Figure 2)

It was revised as follows.

*The model simulation shows that the horizontal distributions of mineral dust concentration at the surface level during the six LRT periods and the vertical distributions on 7 March, when the largest LRT periods occurred during IOP1 (Fig. 4).*

2-5. L200, Did you observe the NaNO3? NaCl can easly react with HNO3 and H2SO4 forming NaNO3 and Na2SO4.

We do not have evidence of NaNO3 in our measurements, although we observed sodium sulfate (original Fig. 7, now Fig 11) and ammonium sulfate (Fig. S8). The possible reasons are as follows. First, nitrate concentration was low, i.e., ~20% of that of sulfate; 0.03-0.12 (nitrate) and 0.15-0.57 (sulfate) µgm-3 (de Sa et al., 2018). Second, nitrate can be volatile and be lost during sampling, storage, and analysis. Third, TEM analysis is insensitive to detect nitrate when it occurs with ammonium sulfate because both include N. Fourth, we measured relatively small sea-salt particles, whereas nitrate prefers to stay with large particles. We added the following explanation in the revised text.

*Nitrate can also react with NaCl and form sodium nitrate. However, we did not observe it in this study because of lower concentrations of nitrate than sulfate during the campaign (de Sá et al., 2018) and insensitivity of nitrate detection in the current TEM analysis.*

de Sá, S. S., et al.: Urban influence on the concentration and composition of submicron particulate matter in central Amazonia, Atmospheric Chemistry and Physics, 18, 12185-12206, 10.5194/acp-18-12185-2018, 2018.

2-6. L223, you observed the 600 nm and >2 um. Recently, Li et al., (2020) found two modes, suggesting different types of biological particles. I am wondering whether the authors observe observed more types with different morphologies. Here the PBA name can be changed to PBAP. PBAP can be more acceptable in many studies such(Després et al., 2012); (Pöhlker et al., 2012); (Li et al., 2020).

We did not find specific bioaerosol particles in the small size. We think fragmented fungal spores (China et al., 2016) contributed to these particles. Although both PBA and PBAP are used for the acronym, we prefer keeping PBA in our manuscript because we termed aerosol species as xx + particle (e.g., sea-salt + particle and mineral + particle). Please find that Pöhlker et al. (2012) uses "primary biological aerosols (PBA)" in their Fig. 4.

*These particles include fungal spores, pollen, bacteria, algae, protozoa, and fragments of*

*plants and organisms (Després et al., 2012; Li et al., 2020) and are known sources of K, Na, P, Cl, and Fe in the Amazon basin (Artaxo et al., 1993; Krejci et al., 2005; Wu et al., 2019). PBA particles >2 μm show unique shapes (Fig. 9), whereas small PBA particles can include fragmented fungal spores (China et al., 2016).*

2-7. L227-229,The diurnal variations can commonly occur in the forests in the worlds. The emission mechanisms could be very complicated depending on winds, RH, and Temperature on plants and soil (Li et al., 2020)(Elbert et al., 2007).

Agree. We revised the sentence as followed.
*The number fractions of PBA particles during the night can be enhanced by high relative humidity (RH), which increases active PBA particle emissions from plants, wind speed, and temperatures (Li et al., 2020; Elbert et al., 2007; Graham et al., 2003; Huffman et al., 2012; Whitehead et al., 2016).*

2-8. L251, Again, what about nitrate.

We added the following sentence here.
*Nitrate can also react with NaCl and form sodium nitrate. However, we did not observe it in this study because of lower concentrations of nitrate than sulfate during the campaign (de Sá et al., 2018) and insensitivity of nitrate detection in the current TEM analysis.*

2-9. L270, Here the authors found 10% sulfate. Maybe need to add one TEM image to show their morphology in Figure 1?

We added a sulfate image in Fig. 1 as well as in Fig. S8.

2-10. L289, Maybe add one TEM image showing the SOA and POA? As the authors found many sulfate particles. The SOA is internally mixed with sulfate as its coating? For example, Li et al. (2016) shows most of SOA coat on secondary inorganic particles in the forest and remote air.

SOA can be a coating over sulfate or others. We added a sentence in Fig. S8 caption. The representative images are shown in Fig. S3, S8, and S9. SOA was also marked in revised Fig. 11.
*Carbon occurs around the sulfate, showing that the sulfates are coated by secondary organic*

*matters and have ns-soot inclusions.*

2-11. L294, why did the authors used biofuel not biomass here?

**During the wet season, there is almost no biomass burning. Instead, local residents burn biofuel for cooking or brick kilns, which contributes to the source of ns-soot.**
*Ns-soot particles that are internally mixed with K-bearing particles can originate from LRT (Saturno et al., 2018) or local biofuel burning for cooking or brick kilns (Martin et al., 2016).*

2-12. L308 other aerosol components
2-13. L309 form various internal mixtures. As Li et al., raise the concept models of mixing structure. The sulfate can be core with SOA coating. They also can be coating on mineral, POA or others.

**The sentence was revised as follows. Please also see our replies to 2-10.**
*Sulfate particles are commonly coagulated or condensed with other aerosol components to form various internal mixtures.*